# Oral Health-Related Quality of Life in Patients Rehabilitated with Dental Implants

**DOI:** 10.3390/healthcare13070813

**Published:** 2025-04-03

**Authors:** Mercy Mora Rojas, Luis Chauca Bajaña, María Rodríguez Tates, Lupe Poussin, Byron Velásquez Ron

**Affiliations:** 1Dentistry School, Prosthesis Department, Universidad de Las Américas (UDLA), Quito 170523, Ecuador; dr_velron@yahoo.com (M.M.R.); lupoussin@gmail.com (L.P.); 2Dentistry School, Periodontology Department, Universidad de Guayaquil (UG), Guayaquil 090613, Ecuador; luischauk@hotmail.com; 3Dentistry School, Universidad de Las Américas (UDLA), Quito 170523, Ecuador; gaby.rodriguezt1@gmail.com; 4Department Prosthesis Research, School of Dentistry, Universidad de Las Américas (UDLA), Av. Colón y 6. Diciembre, Quito 170523, Ecuador

**Keywords:** patient satisfaction, dental implants, dental prosthesis, prosthesis and implants, quality of life

## Abstract

**Background:** A considerable percentage of people in the population have lost their teeth. According to the Word Health Organization (WHO), 40% have lost teeth for multiple reasons. The lack of restorative treatments and the lack of fixed options could impact the quality of life of patients. **Objective:** To determine the quality of life of patients treated with dental implants at the School of Dentistry, University of the Americas, Quito, Ecuador, by the Medical Specialty in Oral Rehabilitation from 2017–2022. **Materials**: The inclusion criterion was patients treated with dental implants age 20 to 70 years old, who agreed to participate in this study, in Quito/Ecuador. The Oral Health Impact Profile (OHIP-14) survey was used. The patients’ clinical history, psychological status, and social status were also recorded. Complementary tests (clinic analysis) were conducted to evaluate the general health status of each patient. **Results:** After the corresponding analysis, the total number of patients who wished to participate in this study in compliance with the inclusion criteria was n = 1303. The seven questions included patient sex, age, and type of prosthesis used in their rehabilitation (single implant, fixed prosthesis on an implant, overdentures, or hybrid prostheses). The obtained results indicated that single implants and overdentures (two jaw implants) improved quality of life, and no statistically significant difference was noted between the sexes. **Conclusions:** The quality of life of patients with total dentures who received dental implants improved substantially, and more fixed dentures helped them to recover their masticatory function appropriately. Single implants were not excluded. In the investigated population, the results were unanimous in how implants improved their comfort and even helped in the recovery of their self-esteem.

## 1. Introduction

The WHO (World Health Organization) has determined that life expectancy has increased in the last 25 years, requiring patients, especially older adults, to comply with activities aimed at improving their quality of life [1]. Previous studies have indicated that patients with complete dentures, owing to their years of use, are already worn and inefficient. In some cases, the bone loss produced in the jaws does not favor the fabrication of a well-fitted or functional denture; nonetheless, the technique used to obtain it does, and this frustrates the patient and creates false expectations that limit their acceptance of new prostheses [2].

The relationship between dental implants and quality of life is a matter of global concern, affecting both high-income and low- and middle-income countries. Tooth loss can negatively impact chewing function, facial aesthetics, and self-esteem, aspects that dental implants seek to restore [3].

One of the factors that influences people’s quality of life is the loss of teeth. Regardless of age, this constitutes complications regarding health and self-esteem [4]. Diseases, such as dental caries and periodontal disease, are categorized as public health problems, and traffic accidents are currently the main source of tooth loss [5]. The use of conventional prostheses, such as removable partial prostheses, fixed prostheses, and complete dentures, has helped the population adequately fulfill their function needs for a long period of time [6,7]. The multidimensional nature of the state of health traditionally focuses on three dimensions: functional capacity, perceptions of social well-being, and aesthetics [7,8]. Multiple instruments have been designed to evaluate the impact that oral diseases have on the population in terms of their quality of life, with a focus on the dental treatments performed [9]. Satisfaction with the service and the efficiency of the procedures performed have also been compared across different types of conventional and nonconventional treatments [10,11]. A high percentage of young patients facing tooth loss currently choose to replace missing teeth with unconventional treatments, such as single implants or fixed prostheses, rather than implants [12]. The elderly population that has maintained removable total dental prostheses currently attends dental clinics to seek implant options that improve the retention of their prostheses [13]. From this point of view, studies have shown that although the quality of life of young patients with implants has significantly improved compared with that of older patients, it is also necessary to consider implant survival and the possible reinterventions necessary to correct pathological situations that have arisen over the years affecting the implant. The diagnosis of this condition can occasionally be extremely complex [14].

The amount of information that patients receive through social networks and advertisements from dental practices provides them with invaluable knowledge about this type of treatment [15]. This has contributed to an increase in the number of patients who proceed with implant treatments. In the present study, patients’ quality of life was evaluated based on the information provided by patients related to oral health (oral health impact profile), as previously mentioned. A multidimensional instrument was used to evaluate the impact of oral problems on aesthetics and functional and psychosocial well-being [16]. Tooth loss is a prevalent condition in Ecuador, especially in older adults, due to factors such as cavities, periodontal diseases, and trauma. Although up-to-date national statistics on edentulism are not available, local studies indicate a significant need for oral rehabilitation. In Ecuador, dental implant treatments are performed in private clinics and universities that offer dental services. The Central University of Ecuador (public), University of the Americas, San Francisco University of Quito, and Two Hemispheres University (Private), for example, have implemented dental implant procedures in their dental clinics, contributing to the training of professionals and community service. There are no precise data on the prevalence of dental implants in the Ecuadorian population. However, the demand for these procedures has increased in recent years, reflecting increased awareness of oral rehabilitation options and an improvement in available dental techniques. Most dental implant procedures in Ecuador are performed in the private sector, which involves high costs that limit access for a large part of the population. The public health system offers basic dental services, but specialized treatments, such as implants, are often not covered [17].

The objective of this study was to determine the quality of life of patients who received dental implants at the University of the Americas in procedures performed by the students of the Medical Specialty in Oral Rehabilitation during the period of 2017–2022.

## 2. Materials and Methods

### 2.1. Data Design and Collection

This was an exploratory, analytical, prospective correlational, and cross-sectional study approved by the Bioethics Committee of the University (CBE170595647/03/24). The samples were obtained by searching the medical records completed in the Oral Rehabilitation Postgraduate Clinic of the University of America in Quito/Ecuador during the period from 2017 to 2022. The total sample defined was 1580 medical records of patients who were treated with dental implant procedures, stored in the archive department of the Faculty of Dentistry Postgraduate Unit of Oral Rehabilitation. The medical records had to have basic information such as mobile phone number or email address; respecting the principle of confidentiality and data management, a database was built for the staff to initiate the contact and follow-up of patients inviting reviews of their treatments, with n = 1303 patient medical records as a definitive sample that met the inclusion criteria

As inclusion criteria, female and male patients who received implant treatment with low, medium, and high levels of complexity such as unitary dental implants, multiple dental implants, over-dentures, and hybrid prostheses were taken into account; the age of the participants was not limited. With the appointment assigned to patients by the archival staff, they were directed to contact the postgraduate students of oral rehabilitation to discuss the objective of this study, and if they agreed and agreed to participate and sign the signed informed consent. The survey was given to them on a physical sheet (physical media), explaining that they could take as long as necessary to complete it. The survey was collected without signatures or identification. Patients who did not wish to participate in this study and patients who did not undergo dental implant rehabilitation were excluded. In this study, only implant rehabilitation was considered, and only quality of life was assessed. Future studies should incorporate questionnaires about fixture operation, operating time, and aesthetic influences.

### 2.2. Research Instrument

The instruments that were considered for the execution of this study were OHIP-14 Spanish Version (Oral Health Impact Profile), which is validated by the World Health Organization in Latin America and South America (Ecuador), and is the most widely used and validated questionnaire to assess the impact of oral health on quality of life, having the original version in English and translations into Spanish, Portuguese, French, German, Chinese, etc. It comprises seven dimensions with two questions per dimension, and assesses the following topics: 1. functional limitations, 2. physical pain, 3. psychological discomfort, 4. physical disability, 5. psychological disability, 6. social disability, and 7. obstacles [3]. The responses were quantified with a Likert scale ranging from 0 to 4 to determine the frequency of the event: 0 never, 1 almost never, 2 sometimes, 3 frequently, and 4 always [4]. GOHAI (Geriatric Oral Health Assessment Index) assesses oral health perception in older adults, with a 12-question questionnaire focused on physical function, pain, and psychosocial limitations, validated for different populations, with the original version in English and translations into Spanish, Portuguese, French, German, Chinese, etc. The OIDP (Oral Impacts on Daily Performance) questionnaire assesses the impact of oral health on the individual’s daily life. It consists of 8 questions in English, and translations into Spanish, Portuguese, French, German, Chinese, etc. It is used in different populations and validated in international studies. CPQ (Child Perceptions Questionnaire) and OHQoL-UK (Oral Health-Related Quality of Life—United Kingdom) were not applicable to this study due to their applicability. The choice of instrument depends on the age group and the objective of this study.

### 2.3. Statistical Analysis

IMB SPSS Statistic 29.0.0 software (Armonk, NY, USA) [18] was used to analyze the data obtained from the proposed questions. Correlations were assessed with Pearson’s test (Chi-squared test), and Spearman’s correlation was used to assess the correlation between the number of prosthetic units and quality of life.

## 3. Results

The results obtained in the present study indicated that more female patients receive implant treatment. From the consultations performed, it was possible to determine that females are much more concerned about their oral health, especially females between the ages of 41 and 50 years, with this group exhibiting greater percentages compared with other age groups (Table 1).

However, the data that are most relevant regarding treatments involve singular implants. In terms of an explanation, when a tooth is lost, the patient does not select a conventional plate or does not want to mutilate their teeth to make a fixed prosthesis. If patients have sufficient bones or undergo bone regeneration, implants are considered the best treatment option. For this reason, the improvement in their quality of life was 65%, a percentage that is considered high. The results obtained for the dimensions* (questions) (Table 2) and question 3 (Q3) for psychological distress based on Pearson’s Chi-squared to determine significant differences (*p* = 0.05, 95% CI) exhibited significant values. This result indicates that the loss of a tooth, especially in the anterior sector, compromises dental aesthetics, influencing the patient’s self-esteem.

In Table 3, for question 4 (Q4), the sample showed significant differences between male and female patients (*p* < 0.05), confirming that the highest percentage was observed in women. For question 5 (Q5), significant differences were obtained between age groups (*p* < 0.05), with a greater percentage of those aged between 41 and 50 years. For the five questions (dimensions), significant differences were noted between the different types of treatments (*p* < 0.05), and patients with individual implants had better quality of life, *p* = 0.000 (*p* < 0.05) (Table 3).

The correlation index between the number of individual prostheses and quality of life was 0.811 (Pearson’s R), and the Spearman’s R of 0.705 indicated a positive correlation (Table 4).

## 4. Discussion

The loss of teeth constitutes an issue for the human body, regardless of the reason for dental loss. This is why it is considered a public health concern by control agencies, and the rates of psychological consequences are considerably high [17]. In some cases, the discomfort generated by conventional dental prostheses has raised even greater concerns about the quality of life of these patients [18]. In 1994, the World Health Organization (WHO) defined quality of life as “the individual’s perception of his or her position in life within the sociocultural context and the value system in which he or she lives with respect to his or her goals, expectations, norms and concerns”. One’s life is related to one’s physical health, psychological state, degree of independence, social relations, and religious beliefs in a complex manner [19]. The option of rehabilitation using dental implants is considered the first option by the population. The preservation of neighboring teeth, as well as the comfort of not feeling the palate or removable prosthetic devices, influence the decision of patients to undergo this type of treatment.

Several studies have shown that females exert greater decision-making power regarding the use of implant restorations [20]. Women exhibit greater self-awareness and concern for their health compared with men, with values exceeding 30%. The oral cavity is considered the main concern because of its function. Life expectancy in females is also a factor to be considered [21] when the use of an implant is considered as a solution to tooth loss.

The authors indicate that individuals who decide to have single or multiple dental implants placed are typically within the age range considered to be active in the workplace. Alhamdan (2023) explained in their study that the average age of implant placement is 35 to 55 years [22], which is consistent with the study results found for participants between 41 and 50 years of age. These results justify the well-being and comfort offered by the treatment and the ability of this group to pay for the treatment.

The studies carried out in university institutions provide a clear idea of what happens with patients in private practice. The experimentation controlled by health institutions generate a shield to people who have received treatments in the care clinics that each university has in its dental clinic care centers, clarifying that the execution of treatments with dental implants are considered clinical procedures that return function, aesthetics, etc. to the patient.

The present study revealed that the hybrid implant technique (“All on 4 and 6”) presented several limitations for the patients, especially in terms of errors when placing the implants (position), fixation selection, economic factors due to extra expenses, and the creation of false expectations with respect to the type of technique to be used. These factors resulted in a significant decrease in patient satisfaction, which contrasts with the findings of Slagter (2021) [23], who reported that the degree of satisfaction due to the use of adequate prosthetic attachments, especially in overdentures, was high. In hybrid techniques, the previously mentioned study indicated that the use of multiple units is considered mandatory for achieving success [24].

Currently, the option of rehabilitation with dental implants is usually considered as the first option by the population. Technological advances allow minimally invasive surgeries to be performed, limiting the percentage of failures to a minimum rate [25].

Taha et al. (2020); Carpentieri et al. (2019) [26,27] reported that the fewer pieces requiring replacement, the lower the probability of complications. This finding aligns with the findings of the present study, which showed that the use of single implants, two implants for multiple fixed prostheses, and overdentures with the O-ring or equator system provided better patient satisfaction.

By improving the internal structure of the implant, the patient can select titanium implant surfaces with and without zirconium oxide [28]. The molecular composition of the implants influences whether the patient feels as if they have metal in their mouth. Patients who adopt homeopathic medicine practices usually discuss these possibilities with their dentist [29,30].

Hartmann et al. (2020) [31] reported that treatments with two implants in older adults improved quality of life and oral health; this is the result of minimally invasive surgeries (guided), the comfort individuals feel given the fixed nature of their prostheses, and even recovery of their self-esteem, which improves their general health. The results obtained in the present study indicate that quality of life improved in older adult patients who received treatment using the overdenture technique.

Similar articles assessing patients’ waiting times indicated that they would have liked to reduce their rehabilitation time. De Siqueira (2022) [32] indicated that the use of immediate loading implants and corresponding previsualization methods adequately improved patient comfort. In total, 77% of those receiving single implants, fixed prostheses, and even dentures were satisfied. It is also recommended that in hybrid prostheses, the ideal outcome is immediate previsualization to obtain better satisfaction rates. Rondone (2024) indicated that despite having performed a bone graft or soft tissue graft, if the primary implant retention is approximately 45 N of torque, he recommends immediate loading in this surgical procedure [33].

In the present study, the percentage of failures was very low, at least during the analyzed period (5 years), including unitary implants and two implants for fixed prostheses or overdentures. However, high quality of life was reported. Compared with the results of the study performed by Al-Asad et al. (2023) [34], a similarly high failure rate was noted after 5 years (51.9%) due to peri-implantitis. However, the main difference between the two studies was sample selection, which considered all implant techniques without exclusion (hybrid). In this case, the level of satisfaction was low.

The importance of evaluating the quality of life of patients receiving any dental prosthetic treatment will always be relevant; however, evaluations using questionnaires could be limited. Occasionally, the respondents omit important information that produces biases in the studies. This is perhaps the greatest limitation that was found in the present study.

The strength of this study was to discover the opinion of patients who received specialty dental care, such as dental implants; the archival departments of public and private universities do not follow-up on patients, which is why this study indicated that they felt abandoned after treatment. The weakness that was considered in this study was the use of questionnaires (validated by the World Health Organization) that for some patients was difficult to understand, which in the pilot tests they tried to limit to control the bias that it could have generated. When treating people of low and medium sociocultural level, the participation and accompaniment of the student was invited, which gave optimal results after the execution of the same.

## 5. Conclusions

The quality of life of patients with total dentures who received dental implants improved substantially. The possibility of their dentures being more fixed helped them to recover appropriate masticatory function. Single implants were not excluded from this study. In the investigated population, the result was unanimous given that implants improved comfort, even aiding in the recovery of patient self-esteem.

## Figures and Tables

**Table 1 healthcare-13-00813-t001:** Demographic patients results.

Age	Physical Condition		Geographical Location			Chronic Diseases		Socio-Economic Status			Chi-Squared (*p*=)	ICE 95%
	Normal	Anormal	Urban	Rural	Peri Urban	Normal	Anormal	Bajo	Medio	Alto		
20/30	100	3	50	20	20	100	3	20	50	20	0.177	
31/40	150	5	100	30		150	5	30	100		0.309	
41/50	190	35	187	24	22	190	35	24	187	22	0.520	
51/60	270	75	283	33		270	75	33	283		0.016	
61/70	150	185	160	10	9	150	185	10	160	9	0.686	
71/80	45	60	34	5		45	60	5	34		0.927	
81/90	10	25	14	2		10	25	2	14		0.301	

**Table 2 healthcare-13-00813-t002:** Improvement in their quality of life.

	Frequency	Percentage %	Total
Quality of Life			1303
Excellent	890	68	
Average	313	24	
Poor	100	8	

**Table 3 healthcare-13-00813-t003:** Statistical data of this study.

Question	Answer				z (≈)				SD				EE (≈)				*p*=				IC 95%			
	Almost Never	Sometimes	Frequently	Never																				
1	326	450	227	110	471	465	481	496	25	35	17	8	0.692	0.969	0.471	0.222	*p* < 0.05	*p* < 0.05	*p* < 0.05	*p* < 0.05	[324.64–327.36]	[448.10–451.90]	[226.08–227.92]	[109.57–110.43]
2	580	463	150	300	465.7	464.4	451.3	470.5	45	36	12	23	1.246	0.997	0.332	0.637	*p* < 0.05	*p* < 0.05	*p* < 0.05	*p* < 0.05	[577.56–582.44]	[461.04–464.96]	[149.35–150.65]	[298.75–301.25]
3	500	420	83	410	476	472	488	477	38	32	6	31	1.05	0.89	0.17	0.86	*p* < 0.05	*p* < 0.05	*p* < 0.05	*p* < 0.05	[497.94–502.06]	[418.26–421.74]	[82.67–83.33]	[408.32–411.68]
4	410	343	140	450	478	476	460	465	31	26	11	35	0.858	0.720	0.304	0.969	*p* < 0.05	*p* < 0.05	*p* < 0.05	*p* < 0.05	[349–471]	[292–394]	[118–162]	[381–519]
5	340	440	73	550	476	472	488	477	26	34	6	42	1.96	1.96	1.96	1.96	*p* < 0.05	*p* < 0.05	*p* < 0.05	*p* < 0.05	[338.59–341.41]	[438.16–441.84]	[72.67–73.33]	[547.71–552.29]
6	480	273		800	468.29	469.26		506.64	37	21		57	1.025	0.582		1.579	<0.000	<0.000		<0.0000	477.99	271.86		796.91
7	420	183			505.36	508.14			420	183			0.831	0.360			<0.000	<0.000			421.63	183.71		
Quality life	390	100	813		585.03	35.94	154.77		20.15	33.33	12.34		0.29	0.36	0.13		*p* < 0.05	*p* < 0.05	*p* < 0.05		[339–451]	[282–384]	[116–170]	

**Table 4 healthcare-13-00813-t004:** Symmetrical measurements.

		Value	Asymptotic Standard Error	T Approximate	Approximate Significance
Interval for Interval	R of Pearson	0.811	0.060	11.441	0.000
Ordinal for ordinal	Correlation of Spearman	0.705	0.074	8.200	0.000
N of valid cases		1303			

## Data Availability

Data are contained within the article.

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
