# Peer review of "Oral Health-Related Quality of Life in Patients Rehabilitated with Dental Implants"

_healthcare, 2025, doi:10.3390/healthcare13070813_

Round 1
Reviewer 1 Report
Comments and Suggestions for Authors
Abstract
-
Please specify the geographic area and the population being studied (e.g., elderly, worldwide, or specifically Ecuador).
-
Correct the year range "20172022" to "from 2017 to 2022" (also apply this correction to line 94).
-
Clarify what is meant by "complementary test."
Introduction
-
Add references to the first paragraph (lines 47–51).
-
Indicate whether the issue discussed is a global problem or primarily affects low- and middle-income countries.
-
Provide a detailed description of the implant situation in Ecuador, including:
-
Epidemiological data.
-
Available treatments.
-
Prevalence of dental implants.
-
Healthcare coverage and treatment options.
-
-
Review commonly used instruments for evaluating oral health-related quality of life (OHRQoL), specifying which are validated and available languages.
Methods
-
Describe how the 1,580 patients were selected for the study, including any sampling procedures or chart review methods.
-
Define "special dental care" in the inclusion criteria:
-
What treatments are included?
-
How complex are they?
-
What age groups are included? Does it encompass both pediatric and adult patients?
-
-
Clarify the meaning of "others were completed by the controls" in line 102. Define the control group.
-
Justify why individuals who did not undergo rehabilitation with dental implants were excluded:
-
What percentage of the sample did this exclusion affect?
-
-
Explain what "OHIP-14 sp" means. Does "sp" refer to a specific language?
-
Specify whether the English or another language version was used.
-
Confirm if the OHIP-14 instrument has been validated in Ecuador.
-
Results
-
Besides age, provide a summary of patient demographics:
-
Age groups, sex, education level, and economic status.
-
-
Clarify the source of the p-values mentioned in lines 142 and 152, as Table 1 does not present any p-values.
-
Report the actual p-values rather than using "p < 0.05."
-
Explain the values mentioned in lines 157–160, as they are not present in Table 2.
-
Correct "cant" to "count" in Tables 2 and 3.
-
Verify if Table 2 questions are domain-based (e.g., D1–D7). Explain how 14 questions measure these domains.
-
Consider using a multivariable model for the outcome "quality of life," adjusting for covariates. Correlation and bivariate analyses may not sufficiently capture the relationship between implants and quality of life.
Discussion
-
Address how study results from the university setting can be generalized to the broader population.
Author Response
REVIEW 1 .
- Please specify the geographic area and the population being studied (e.g., elderly, worldwide, or specifically Ecuador).
Corrected according to the suggestion of the reviewer
Correct the year range "20172022" to "from 2017 to 2022" (also apply this correction to line 94).
Corrected according to the suggestion of the reviewer
- Clarify what is meant by "complementary test."
Corrected according to the suggestion of the reviewer
Introduction
- Add references to the first paragraph (lines 47–51)
Added according to the suggestion of the reviewer
Indicate whether the issue discussed is a global problem or primarily affects low- and middle-income countries.
Added according to the suggestion of the reviewer
- Provide a detailed description of the implant situation in Ecuador, including:
- Epidemiological data.
- Available treatments.
- Prevalence of dental implants.
- Healthcare coverage and treatment options.
Added according to the suggestion of the reviewer
- Review commonly used instruments for evaluating oral health-related quality of life (OHRQoL), specifying which are validated and available languages.
- Corrected and added according to the suggestion of the reviewer
Methods
- Describe how the 1,580 patients were selected for the study, including any sampling procedures or chart review methods.
Corrected and added according to the suggestion of the reviewer
- Define "special dental care" in the inclusion criteria:
- What treatments are included?
- How complex are they?
- What age groups are included? Does it encompass both pediatric and adult patients?
- Clarify the meaning of "others were completed by the controls" in line 102. Define the control group.
Corrected and added according to the suggestion of the reviewer
- Justify why individuals who did not undergo rehabilitation with dental implants were excluded:
- What percentage of the sample did this exclusion affect?
Corrected and added according to the suggestion of the reviewer
Explain what "OHIP-14 sp" means. Does "sp" refer to a specific language?
- Specify whether the English or another language version was used.
Corrected and added according to the suggestion of the reviewer
- Confirm if the OHIP-14 instrument has been validated in Ecuador.
Corrected and added according to the suggestion of the reviewer
Results
- Besides age, provide a summary of patient demographics:
- Age groups, sex, education level, and economic status.
Corrected and added according to the suggestion of the reviewer
- Clarify the source of the p-values mentioned in lines 142 and 152, as Table 1 does not present any p-values.
Corrected and added according to the suggestion of the reviewer
- Report the actual p-values rather than using "p < 0.05."
Corrected and added according to the suggestion of the reviewer
- Explain the values mentioned in lines 157–160, as they are not present in Table 2.
Corrected and added according to the suggestion of the reviewer
- Correct "cant" to "count" in Tables 2 and 3.
Corrected and added according to the suggestion of the reviewer
- Verify if Table 2 questions are domain-based (e.g., D1–D7). Explain how 14 questions measure these domains.
Corrected and added according to the suggestion of the reviewer
- Consider using a multivariable model for the outcome "quality of life," adjusting for covariates. Correlation and bivariate analyses may not sufficiently capture the relationship between implants and quality of life.
Corrected and added according to the suggestion of the reviewer
Discussion
- Address how study results from the university setting can be generalized to the broader population.
Corrected and added according to the suggestion of the reviewer

Reviewer 2 Report
Comments and Suggestions for Authors
Thank you for allowing to review your work. I have following feedback if you may consider it:
1- Introduction needs research need gap and what is the novelty of present study and how will it inform the future direction
2-Lines47-51 requires citation for the claim made. If it’s authors personal opinion, - please consider stating that
3-Lines 81-84 not clear in the context of line prior and line following the sentence. Please clarify if the manuscript develops and tests the instrument or is it using already developed instrument and cited study developed it.
4-Lines 123 software needs appropriate citation and release version details
5-Results section- the patient demographic descriptive results table is missing. i.e., patient physical status/condition, chronic conditions, age, geography etc, provide 95% CI for estimates and significance level. Without it the table results seem to provide biased information from statistical study standpoint of view as several underlying diverse patient characteristics might affect quality of life . It is important in establishing study validity threats including sample selection/bias etc. please consider adding a table of descriptive statistics for included sample
6-Table 2 and 3 please provide 95% CI for each dimension question to determine if the estimate CI is broader or narrower and for audience to establish the effect accuracy on its basis
7-In discussion section please consider mentioning study strength and limitations. In limitations section please consider identifying unaccounted confounders that might affect study validity threats. Lastly authors should acknowledge that correlation is not directly association or causal relation conclusive unless all the confounders and selection bias issues are accounted for in the statistical analysis for being transparent with readers.
In general from statistical standpoint of view the present manuscript is not strong. Unless made significant changes I do not vote towards publication as it doesn’t meet journal standards. ​Thank you!
Author Response
Review 2
- Thank you for allowing to review your work. I have following feedback if you may consider it:
- 1- Introduction needs research need gap and what is the novelty of present study and how will it inform the future direction
Corrected and added according to the suggestion of the reviewer
- 2-Lines47-51 requires citation for the claim made. If it’s authors personal opinion, - please consider stating that
Corrected and added according to the suggestion of the reviewer
- 3-Lines 81-84 not clear in the context of line prior and line following the sentence. Please clarify if the manuscript develops and tests the instrument or is it using already developed instrument and cited study developed it.
Corrected and added according to the suggestion of the reviewer
- 4-Lines 139 software needs appropriate citation and release version details
Corrected and added according to the suggestion of the reviewer
- 5-Results section- the patient demographic descriptive results table is missing. i.e., patient physical status/condition, chronic conditions, age, geography etc, provide 95% CI for estimates and significance level. Without it the table results seem to provide biased information from statistical study standpoint of view as several underlying diverse patient characteristics might affect quality of life . It is important in establishing study validity threats including sample selection/bias etc. please consider adding a table of descriptive statistics for included sample
Corrected and added according to the suggestion of the reviewer
- 6-Table 2 and 3 please provide 95% CI for each dimension question to determine if the estimate CI is broader or narrower and for audience to establish the effect accuracy on its basis
Corrected and added according to the suggestion of the reviewer
- 7-In discussion section please consider mentioning study strength and limitations. In limitations section please consider identifying unaccounted confounders that might affect study validity threats. Lastly authors should acknowledge that correlation is not directly association or causal relation conclusive unless all the confounders and selection bias issues are accounted for in the statistical analysis for being transparent with readers.
- Corrected and added according to the suggestion of the reviewer
- In general from statistical standpoint of view the present manuscript is not strong. Unless made significant changes I do not vote towards publication as it doesn’t meet journal standards. ​
- Corrected and added according to the suggestion of the reviewer
- Thank you!

Round 2
Reviewer 2 Report
Comments and Suggestions for Authors
Thank you for your consideration.